# Exploration of a workflow for the classification and identification of co-purified protein complexes yields new structures and multiple MSP assembly states

Qingyang Zhang[1,2], Abhinandan Venkatesha Murthy[3], Carsten Mim[1]*

1 Department of Protein Science, Kungliga Tekniska Högskolan (KTH); Stockholm, Sweden,
2 Department of Biomedical Engineering and Health Systems, Kungliga Tekniska Högskolan (KTH); Stockholm, Sweden, 3 Institute of Biotechnology, University of Helsinki, Helsinki, Finland

* carmim@kth.se

## Abstract

Native protein complexes have garnered interest as targets for structural dissemination. Cryogenic electron microscopy (cryo-EM) with its ability to image protein mixtures is the most promising tool to enable structural proteomics. Additionally, image processing has evolved and can deal with conformational and compositional heterogeneity. Integrative approaches, namely mass spectrometry in conjunction with cryo-EM, have made it possible to characterize and identify complex mixtures. However, this comes at a cost of generating models and interpreting mass spectra. Here we present a modified approach that builds on publicly available software. By generating maps around 4 Å and unsupervised model building we were able to identify the most abundant proteins in our sample. This sample consisted of co-purified membrane proteins in nanodiscs. We found a novel structure and unexpected nanodisc assemblies. Our maps imply a direct interaction between membrane proteins and membrane scaffolding proteins.

## Introduction

Visualization of native protein complexes is essential to identify the physiological composition and stoichiometry of biologically relevant protein complexes. In some cases, native complexes show that structures and conformations solved with heterologous expressed proteins are not representative [1]. Protein concentration and heterogeneity of native samples too often restricts structure determination by X-ray crystallography or NMR. However, Cryogenic electron microscopy (cryo-EM) has allowed for reconstruction of native complexes early on [2]. But only the advent of single particle analysis (SPA) started a new era in cryo-EM and made it routine to solve structures with a resolution high enough to build models *de novo* [3]. Over the last 5 years, the deposition of structures solved by SPA cryo-EM has increased

**Data availability statement:** CryoEM maps were deposited at the Electron Microscopy Data Bank (https://www.ebi.ac.uk/emdb/) with the accession codes EMD-52452 (ACRB), EMD 52451 (BO3) and EMD-52450 (ARNC). The models were deposited at the Protein Data Bank (https://www.wwpdb.org/) with the accession codes 9HWL (ACRB), 9HWK (BO3) and 9HWJ (ARNC).

**Funding:** This study was supported by the Carl Tryggers Stiftelse för Vetenskaplig Forskning grant CTS 21:1630 (CM). The funders had no role in study design, data collection and analysis, decision to publish, or preparation of the manuscript.

**Competing interests:** The authors have declared that no competing interests exist.

almost exponentially (https://www.ebi.ac.uk/emdb/statistics/emdb_entries_year). It has been shown that cryo-EM can be a fast structural method, by generating the high resolution structure of a plant ribosome, from the leaf to the final map, in one day [4]. Ever since, structural biology of native samples has become an expanding field [1,5–7]. Maximum likelihood and deep learning methods have been used successfully to deal with sample heterogeneity, in particular compositional heterogeneity and conformational heterogeneity (e.g., RELION, CryoDRGN, cryoSPARC [8–10]). This positions cryo-EM as an ideal tool for structural proteomics. One early approach is based on a shotgun proteomics, by fractioning cell lysates through size chromatography and subsequent cryo-EM/negative stain EM characterisation of 2D classes and 3D maps [11,12]. This approach only works with proteins of high natural abundance, e.g., housekeeping or structural proteins. Therefore it is necessary to enrich low copy proteins, e.g., by density centrifugation [4,7] or by using an affinity chromatography resin with a broad specificity [13]. Immobilized metal affinity chromatography (IMAC) has the ability to co-purify a variety of proteins in *E. coli* [14]. This feat has been used to structurally investigate co-purified complexes [15]. However, structural proteomics for membrane proteins represents a unique challenge, because membrane proteins require stabilization, most often with detergents. Lipid nanodiscs, which are a shell of stabilised lipids around a protein, offer an opportunity to preserve the lipid environment around the protein. The most common nanodisc system uses the membrane-scaffold-protein (MSP) technology and is based on the apolipoprotein A (ApoA) protein, yet other systems are gaining popularity (e.g., reviewed in [16]). A recent study combined a database search of cryo-EM-based, *de novo* models with mass spectrometry to identify heterogeneous protein populations. This approach has proven to be a powerful tool [13,17]. However, this workflow may become tedious because the models must be built, although automated solution for model building exist [18]. Additionally, mass spectrometry data collection and interpretation requires expertise. Lately, deep-learning-assisted model building has opened the avenue for unsupervised model building and protein identification [19–21].

Here, we present a workflow for the identification of cryo-EM maps from a heterogeneous protein mixture. We show that particle classification and unsupervised model building may be sufficient to characterize complex samples with fewer resources. We were able to identify and build models for 3 membrane proteins co-purified from IMAC, one of them represents a new structure. These identified proteins were confirmed with MS to validate our approach. These proteins were reconstituted in MSP derived nanodiscs which exhibited unusual assembly states. Namely, smaller nanodiscs and non-circular shapes. Our models allowed us to speculate that proteins sidechains may act as interaction sites between membrane proteins and the scaffold proteins.

## Results

### Workflow for the preparation of co-purified membrane proteins in nanodiscs

Initially, this study focussed on the production of a mammalian membrane protein in *E. coli*. Xiang *et al.* demonstrated that the application of an osmotic shock in addition

to a cold shock increased the expression of a pentameric ion channel [22]. Therefore, we adapted this protocol with the goal to express the human Zinc activated ion channel (ZANC). After lysis and removal of debris and cytosolic components, we solubilized the membrane with a high salt buffer (0.9 M KCl) to minimize unspecific, ionic interactions. The solubilized membrane proteins were enriched with immobilized metal affinity chromatography (IMAC). Because we were interested in lipid-protein interactions, the enriched protein was incorporated into MSP variant 2N2 (MSP2N2) nanodiscs, as described for other pentameric channels (e.g., [23]). The sample was further processed with size exclusion chromatography (SEC) and fractions were pooled for characterization. SDS-PAGE showed multiple proteins (Fig 1). IMAC is known to enrich native *E. coli* proteins (e.g., [14,15]).To characterize these proteins we performed mass spectrometry of the concentrated SEC fractions. We were able to identify many *E. coli* specific proteins in our sample (S2 Table). Yet, without quantification, the large number of hits makes an identification of the most abundant bands challenging. We decided to use a cryo-EM based approach to assist us with the identification, which had been used in the past by others [17]. In contrast to the latter study, we wanted to build the model unsupervised to assist us with the identification (Fig 2 for the general workflow). The ModelAngelo package [19] uses a hidden Markov models (HMMs) search to query the sequence [20] and demonstrated a better performance than FindMySequence [21] to identify proteins.

## Unsupervised model building of cryo-EM maps can identify proteins

ModelAngelo was reported to build cryo-EM maps with a resolution of 4 Å and better [19]. Notably, about 75% of all cryo-EM maps deposited in 2024 are 4 Å and better (https://www.ebi.ac.uk/emdb/statistics/emdb_resolution_year: retrieved 12/2024), which encouraged us to proceed with our approach. We collected almost 18,000 movies. All processing was performed in cryoSPARC [24]. To get an idea of the most prominent proteins in our mixture, we picked particles manually from the motion corrected micrographs. The picked particles were classified in 2D to generate crude templates for template picking. Initially, this generated about 3 million particles and we performed 2D classification to obtain 200 classes (Fig 2, S2, S4 Figs). Among the 200 classes we suspected 3–4 different proteins, which may have been biased by the choice of the template. As a control, we used featureless, circular templates with a diameter of 80–180 Å ('blobs'), and we uncovered the same proteins after an additional round of 2D classification (S2 Fig). We proceeded with separating particles for proteins we deemed similar and applied further rounds of 2D classification and an *ab initio* reconstruction with multiple classes for each particle set. The *ab initio* maps were refined with Non-Uniform Refinement [25]. This approach yielded in one cryo-EM map for each different particle set. To improve the particle set we used the best 2D classes of each individual map as a template for picking particles again. After motion correction of the 'cleaned up' particles, we obtained 3 maps with an overall resolution of <4 Å, although the local resolution may be lower (Fig 3). Based on previous studies [26,27] and visual inspection of the cryo-EM map, Cytochrome bo(3) ubiquinol oxidase (BO3) was identified as one of the proteins (S3 Fig). We chose this as the first map for modelling by ModelAngelo. We did not supply an individual sequence, instead we supplied the whole *E. coli* proteome and performed an HMM search. Apart from the non-protein moieties, all subunits were identified (S3 Table for all hits, S6 Table). To assess the quality of each modelled fragment and thus the identity of the protein, we inspected the HHMer output carefully for all hits, specifically the E-value, which represents a probability for the sequence match. Because the ModelAngelo output is an ensemble of fragments, we used a publicly available model (pdb ID: 7N9Z, [26]) to model our map manually, for final refinement and deposition of the refined model. Even without refinement the backbone trace of unsupervised ModelAngelo model (Fig 3A, magenta), the manually refined model (Fig 3A, cyan) and the unsupervised model are in good agreement. Next, we turned to the protein with the largest map. It is an asymmetric trimeric protein and was identified as the multidrug efflux pump subunit ACRB (Fig 3B, S4 Fig, S4 Table, S6 Table). Again, we used a previously deposited model (pdb ID: 2HRT, [28]) for manual modelling, refinement and deposition of this model. The unsupervised model (Fig 3B, magenta) and the manually refined model (Fig 3B, cyan) were in good agreement, again. The last map presented a challenge because it represented the smallest protein with a large variation in resolution and areas worse than 4 Å in resolution. Nevertheless, ModelAngelo identified the protein

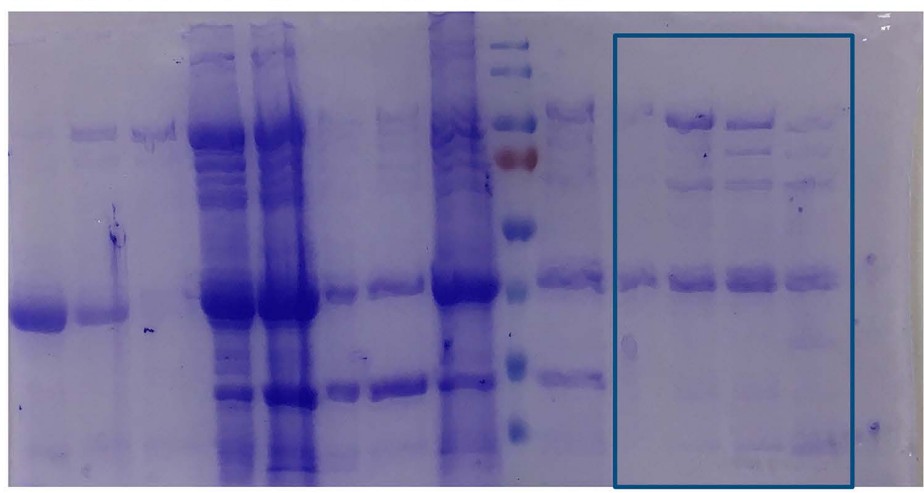

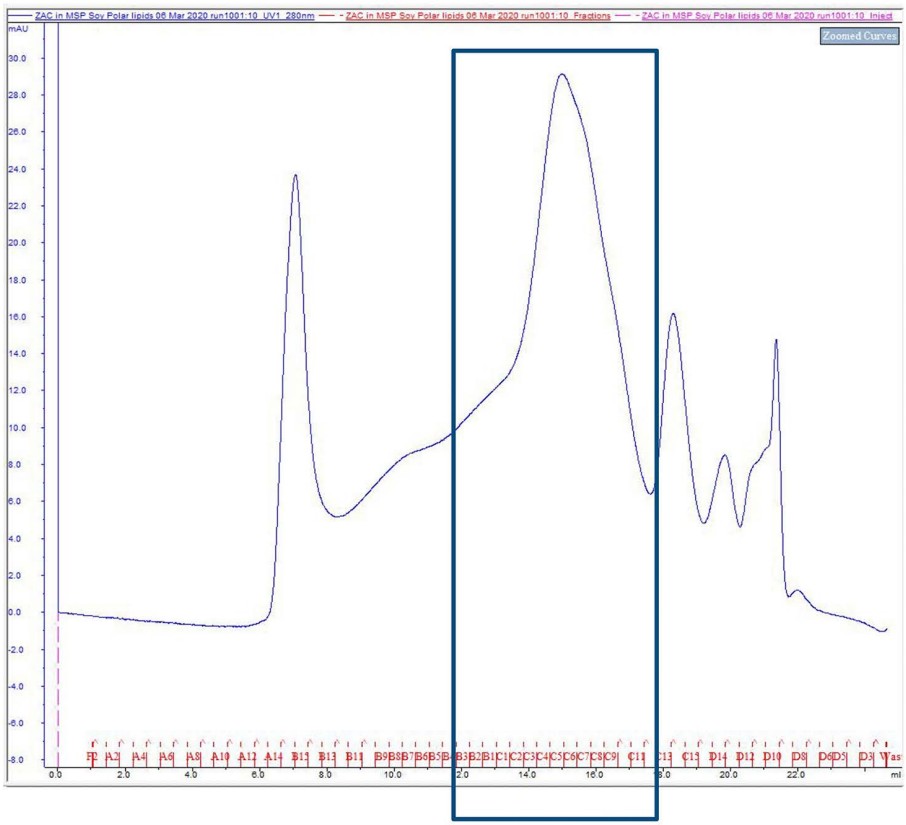

**Fig 1. Sample preparation Top: SDS-PAGE for samples of each step: FT: flow through of the Ni-NTA resin; W: Wash, E1-5: Elution fractions per 1 resin volume; B: Ni-NTA-resin beads after elution; M: marker; I: concentrated protein embedded MSP2N2 nanodisc sample for the SEC run; C3-14; SEC elution fractions.** Bottom: SEC profile of the protein embedded MSP2N2 nanodiscs the box denotes the fractions used for concentrating the sample.

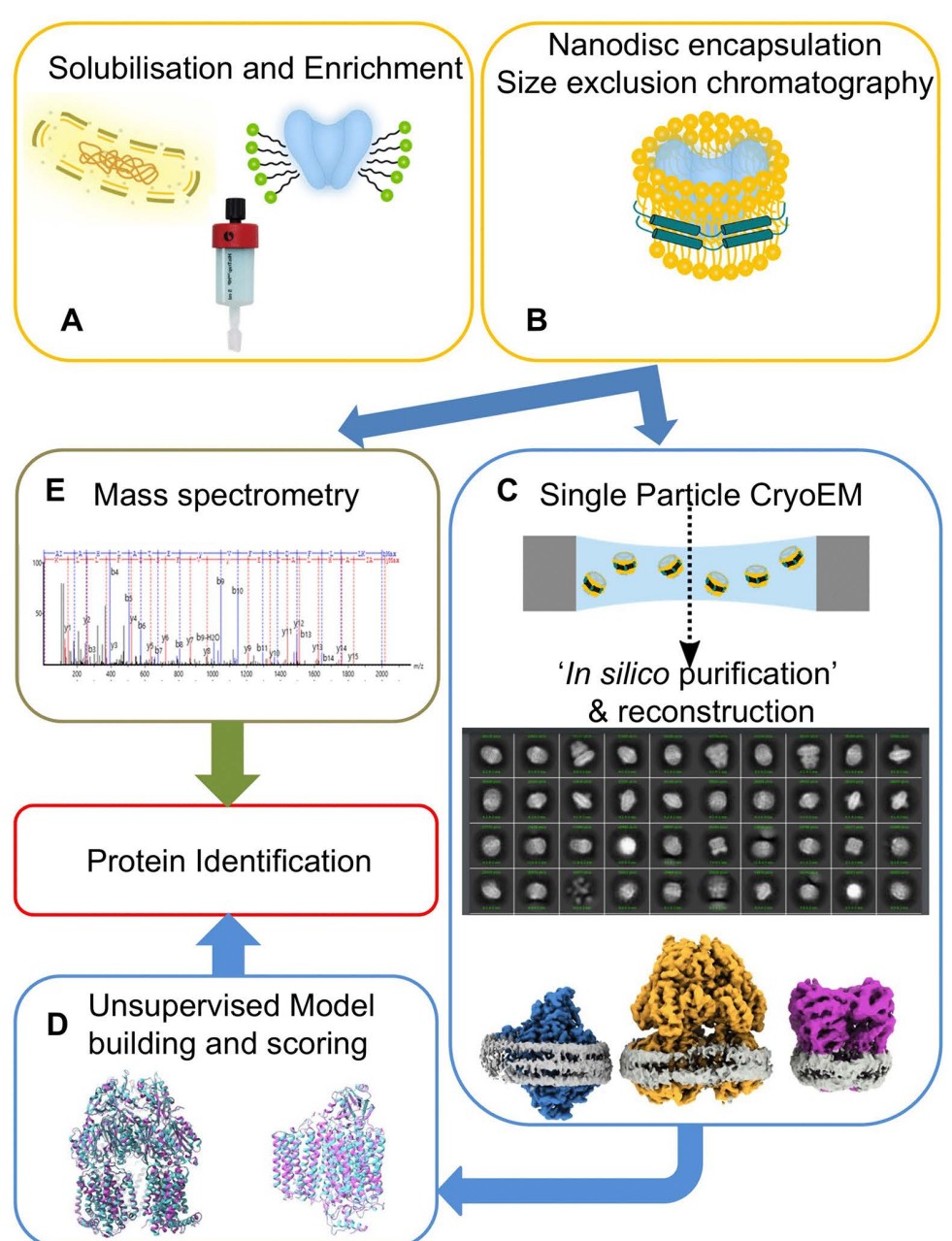

**Fig 2. Workflow for the production and *in silico* identification of membrane proteins (A)** *E. coli* **membrane proteins are solubilized and enriched by IMAC (B) the enriched proteins are then embedded into nanodiscs with the same scaffolding protein (MSP2N2) and separated by size exclusion chromatography (C) the protein mix is then imaged by single particle cryo-EM.** The particles are classified and reconstructed separately. **(D)** The best maps are used to build a model with ModelAngelo [19] without sequence input to generate hits (S3-S6 Tables). In addition, a manual search was performed to identify proteins. **(E)** the imaged sample was analysed via mass spectrometry for validation.

as Undecaprenyl-phosphate 4-deoxy-4-formamido-L-arabinose transferase (ARNC, S5 Table, S5 Fig). Because this is a new structure, we used the AlphaFold model (ID: P77757) for refinement. Generally, the unsupervised model (Fig 3C, magenta) and the manually refined model (Fig 3C, cyan) agreed well. However, the quality of the model was problematic,

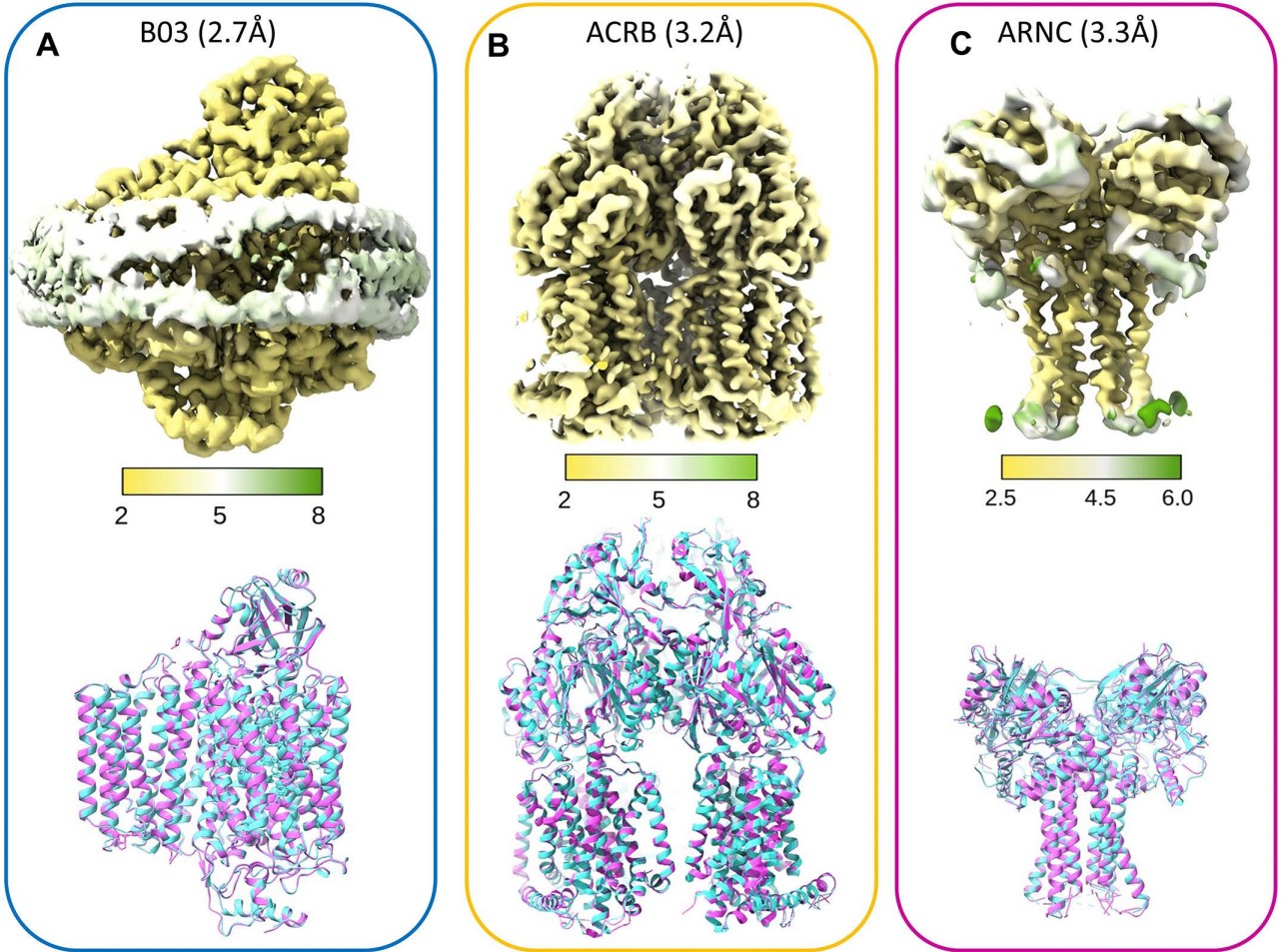

**Fig 3. Identification and model building of the reconstructed maps.** (A) top: local resolution for the Cytochrome bo(3) ubiquinol oxidase subunit 4 map in Å with an overall resolution of 2.7 Å. The density for the membrane scaffolding protein is shown as well. The comparison of the backbone trace between the manually refined model (cyan) and the ModelAngelo model (magenta). (B) top: local resolution of the map for Multidrug efflux pump subunit ACRB in Å with an overall resolution of 3.3 Å. bottom: The comparison of the backbone trace between the manually refined model (cyan) and the ModelAngelo model (magenta). (C) top: local resolution of the protein density for Undecaprenyl-phosphate 4-deoxy-4-formamido-L-arabinose transferase (ARNC) in Å with an overall resolution of 3.3 Å. bottom: The comparison of the backbone trace from the manually refined model (cyan) with the unrefined model build by ModelAngelo (magenta) shows little deviation. The deposited models for BO3 and ACRB were based on pdb ID: 7N9Z pdb and ID: 2HRT respectively. The deposited model for ARNC was based on AlphaFold (ID: P77757).

because some of the assigned sidechains (e.g., S6A Fig) and connectivity between some of the helices was incorrect. In the ModelAngelo/HMMer output we recognized these issues, when we compared the E-values for the B03 model with the E-values from the ARNC model, with the latter being higher (S3 Table vs S5 Table, S6 Table). We were able to verify the identity of all proteins by mass spectrometry (S2 Table) to validate our approach.

**The nanodiscs show various topologies and point to putative interaction sites of the membrane proteins with the scaffold**

We solved the structures of 3 different membrane proteins that were reconstituted in MSP2N2 nanodiscs (Fig 4). With the CHARMM-GUI [29] we built a model for a MSP2N2 nanodisc. The theoretical diameter of this nanodisc is 165 Å,

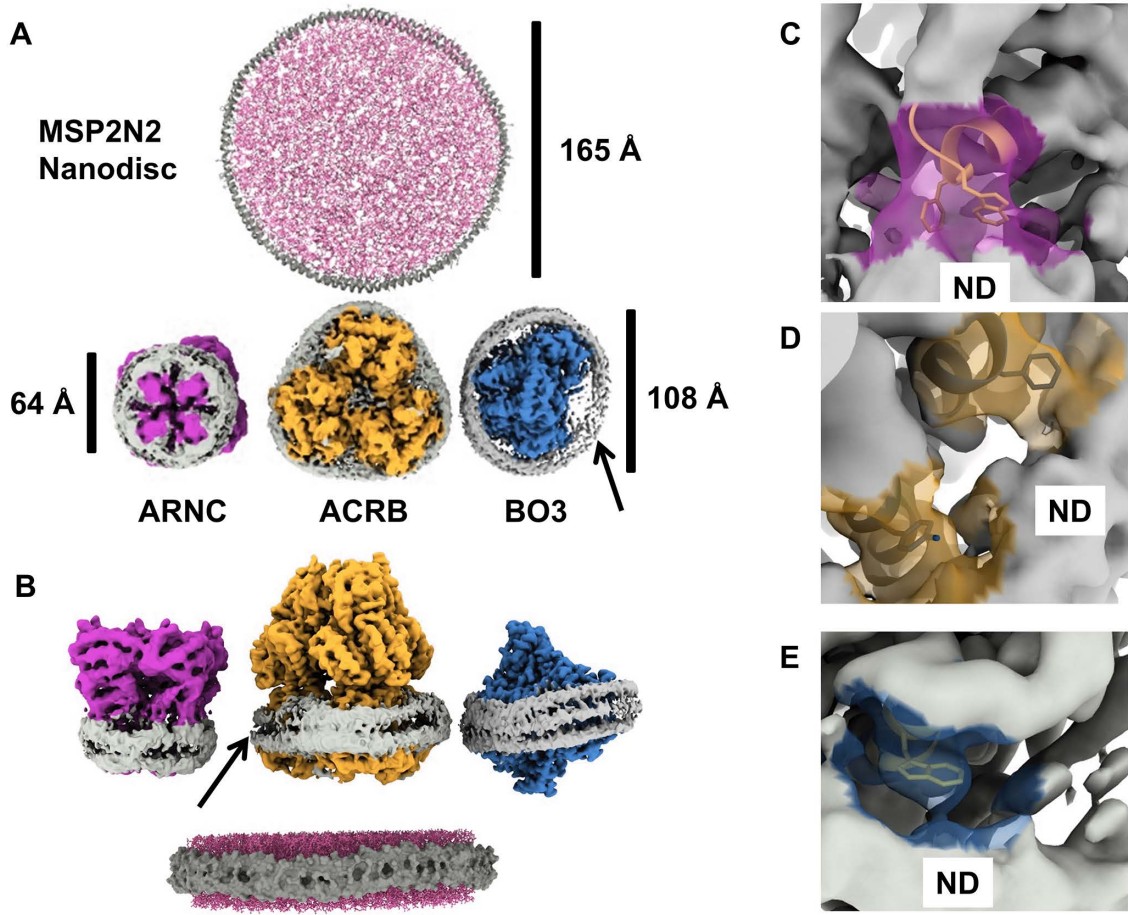

**Fig 4. Properties of the nanodisc embedded membrane proteins.** (A) top: Top view of a computationally assembled nanodisc formed by MSP2N2 (grey) and lipids (pink) by the CHARMM-GUI [29] with a diameter of 165 Å and bottom: the nanodiscs found in samples. Unsharpened maps of the identified proteins show a different diameter and shape of the nanodisc. The ARNC (magenta) nanodisc (grey) has a diameter of around 65 Å. The largest nanodiscs formed around ACRB (yellow) and BO3 (blue) and similar (108 Å) but their shape is different. We measured the distance by placing markers at the edge of the nanodisc density. (B) side view of the structures in **A.** The scaffolding protein (MSP2N2) in the ideal nanodisc is represented as a space filling model (grey) around lipids (pink). The hydrophobic core is covered by two adjacent MSP2N2 helices (bottom). ARNC and ACRB nanodiscs have densities resemble the two-helix assembly seen in MSP2N2-only nanodiscs. BO3 exhibits three rungs of density to form a nanodisc. The presumed scaffold protein density appears to be weakest in areas without the membrane protein in the ACRB and BO3 maps (arrows). (C) putative interaction sites between ARNC (transparent purple) and the assumed MSP2N2 density (ND). Residues W165 and F136 (yellow) are within 5 Å of the assumed MSP2N2 density. **(D)** ACRB (transparent yellow) contains residues (Y554 and F918) that are within 5 Å of the assumed MSP2N2 density (grey). (E) one putative interaction site in BO3 (light blue) showing W34 within 5 Å of the assumed MSP2N2 density (grey).

consistent with the size of an empty nanodisc [30]. The computationally generated nanodisc was substantially larger than any of the distances measured in the experimental nanodiscs (ARNC = 65 Å, ACRB = 108 Å, BO3 = 108 Å). We measured the distances by manually placing markers in the density. This size discrepancy between the empty and occupied nanodiscs has been observed in other cryo-EM maps of MSP2N2 encased proteins. Those nanodiscs were 100–120 Å in diameter [31–33]. The computationally assembled MSP2N2 nanodisc was framed by two helices (Fig 4 B bottom). In our map we saw variation of this assembly. While the ARNC and ACRB nanodiscs showed two density rungs, some positions of the BO3 nanodisc had 3 density rungs, reminiscent of the assembly observed by Roh *et al* [34]. Notably, we saw that the density around the nanodisc is strongest close to the membrane protein (Fig 4A and 4B, arrows at zones with lower

density). Therefore, we hypothesized that there is an interaction between MSP2N2 and the protein it encapsulates. We searched for residues in our models that are within 5 Å of the presumed MSP density. 5 Å was chosen, because it is slightly longer than the contour length of amino acid sidechains [35]. We identified several non-polar and aromatic side-chains in each protein (Fig 4C-4E, S4 Fig). In ARNC, a pair of aromatic residues (W135, F136 Fig 4C) is situated near the presumed MSP density. ACRB shows more potential contact sites through aromatic residues (Y554 and F918, Fig 4D; W515, S6E Fig) and an arginine (R540, S6E Fig). Last, in BO3 we identified a pair of tryptophan side chains (W34, Fig 4E, W454 S6F Fig) and an arginine (R455, S6F Fig) close to the presumed MSP density. We could not generate a well-resolved map for the MSP2N2 density itself, that would enable us to build a model. Yet, all these putative contact sites enforce the idea that membrane proteins have a more active role in the nanodisc assembly.

## Structural features of the identified protein complexes

Our purification protocol used high ionic strength to assist solubilisation. Also, we embedded bacterial proteins into soy lipids. Therefore, we were interested if the structural features of the discovered proteins are altered. In the BO3 map we observed non-protein densities at the position where we would expect lipids (Fig 5A, yellow). Most of the densities were not defined enough to build models for specific lipids. However, we could fit some of the lipids seen in a deposited model

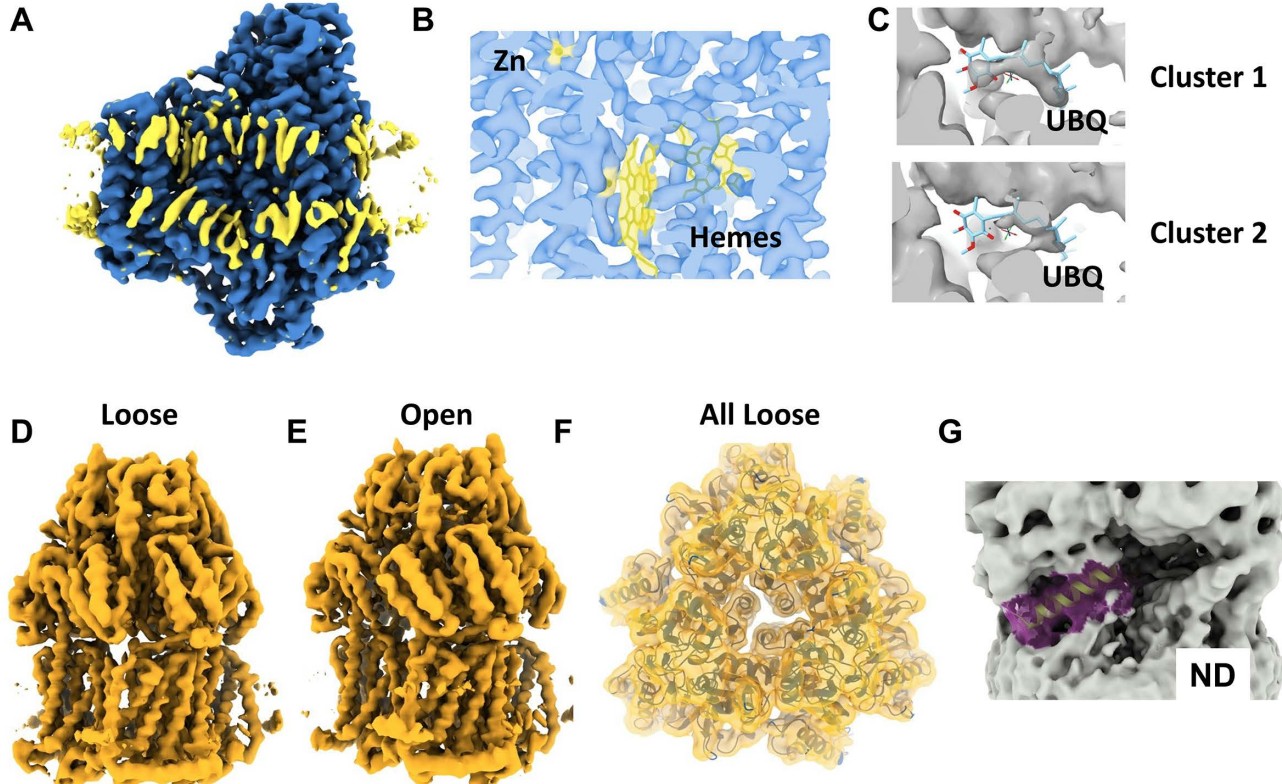

**Fig 5. Structure-function features of the characterized proteins (A) Non-protein densities (yellow) in the BO3 map at the position of the bilayer's hydrophobic core. (B)** Density for the protein (transparent blue) and non-protein moieties (transparent yellow) in BO3. **(C)** Density for 2 of the 3 3DVA cluster maps (grey) at the position of the electron carrier ubiquinone (UBQ, stick representation) in BO3. In all 3 clusters the UBQ-associated density varied. **(D)** and **(E)** 3DVA analysis of the ACRB dataset (See S1 Movie) shows several conformations of the efflux pump, 'Loose', 'Open'. The 'Tight' conformation is not shown here for simplicity but modelled. **(F)** one of the clusters from the 3DVA shows an all 'Loose' conformation **(G)** Putative substrate binding domain of ARNC (yellow in purple envelope) and its proximity to the membrane.

(S6C and S6D Fig) [26]. In addition, we were able to model the functional groups of the electron transfer pathway, namely the heme moieties and the coordinated metals $Zn^{2+}$ and $Cu^{2+}$ (Fig 5B). Li *et al* reported that the binding site for the electron carrier ubiquinone (UBQ) is dynamic. Therefore, we performed a 3D variance analysis (3DVA) for the BO3 dataset [10]. The overall structure showed little flexibility, but we observed that the presumed UBQ density varied (Fig 5C). This could either indicate that some particles lost UBQ or that the position of UBQ is not fixed. In the previous section we showed that all proteins have side chains that are near the proposed nanodisc MSP density. There have been concerns that nanodiscs may change the structure of the embedded membrane proteins [36,37]. ACRB is a pump whose subunits cycle through 3 conformations: 'loose' (L) and 'tight' (T) are structurally similar and have substrate binding sites open to the periplasm; 'open' (O) has the substrate binding site closed to the periplasm [38]. We performed a 3DVA with the ACRB particle set. When we separate 4 clusters, we observe that 3 clusters are in an asymmetric LTO state (Fig 5D for the L protomer conformation, and 5E for the O protomer conformation, the T state was omitted for simplicity, S1 Movie M1) [28]. The LTO state was modelled for our reconstruction (Fig 3B). Yet, one 3DVA cluster represents a symmetric state that we believe is in an all 'loose' state, based on the fitting of the 'loose' subunit. Last, we solved the structure of ARNC embedded in the membrane, unlike the structures of homologues, which were solved in detergent [39,40]. The helices closest to the membrane (amino acids: 215–228 and 135–153) are both amphipathic helices (S4B Fig) and the map suggest a placement of these helices in one leaflet. These helices comprise the membrane adjacent part of the substrate binding pocket in ARNC homologues [39,40].

## Discussion

Structural biology has long relied on pure samples with engineered proteins that enhance stability to either limit aggregation at high concentrations or to endure the long timescales necessary to grow crystals. Because high resolution cryo-EM images proteins complexes/particles directly, sample requirements are less restrictive. As a result, solving structures of native proteins and complexes has become more common (e.g., [1,4,12,41]). However, some of studies still require the use of affinity proteins like nanobodies. The 'build and retrieve' approach worked with an IMAC resin, yet supervised (*de novo*) model building or *a priori* knowledge of the reconstructed map is part of that workflow [17]. Here we present an alternative for an unbiased workflow to identify reconstructed maps of unknown proteins, if the maps are in the common resolution range for single particle cryo-EM (~4 Å). Our workflow suggests that mass spectrometry is not strictly necessary, which makes our approach more accessible because it reduces the cost and time for mass spectrometry data collection and interpretation. We have not tested the workflow for a map with less than 4 Å in resolution, but the ARNC map has many areas where the resolution is less than 4 Å and the assignment of residues in some areas is error prone. Yet, there were enough modelled fragments that allowed for a correct identification. We imagine that the identification and differentiation of protein isoforms may prove difficult at lower resolutions. In this case the close inspection of the modelling statistics (E-value, score and bias) will be necessary as well as the inspection of the modelled amino acids. Our workflow may also be problematic for low abundance proteins, especially if they have a similar shape as more abundant proteins. The latter risks a wrong classification of particles. Last, if one is interested in post-translation patterns or ligands, cryo-EM at lower resolutions may not correctly identify ligands. For instance, we identified lipids the BO3 map, yet their identity could not be determined by the map alone. Ultimately, mass spectrometry may help in those cases.

Notably, the combination of proteins we identified has been unique. BO3 commonly co-purifies with IMAC [17,26,27]. However, ACRB and ARNC are novel proteins appearing as co-purified proteins. While ARNC represents only a small fraction of the particles, ACRB is a prominent component. This may be due to the chosen templates but even 'blob' based picking resulted in a population of ACRB/ARNC in the 2D classification (S2 Fig). It appears that each 'non-specific enrichment' purification may result in different protein populations and may depend on the ionic strength, used detergent, expression protocol and other factors. Interestingly, ACRB and ARNC are both proteins that are overexpressed in response to antibiotics.

Our study also provided insight into interactions and assembly of MSP based nanodiscs. We demonstrate MSP2N2 can accommodate different nanodisc sizes and shapes. For the ARNC map we suggest that one MSP2N2 molecule may be enough to form a nanodisc because the nanodisc is so small. This deviates from the common assumption that 2 MSP molecules per nanodisc are needed [42]. Although it has been reported that 3 MSP molecules can form a nanodisc [43]. However, the smaller size of the BO3 nanodisc makes this unlikely, despite the observation of 3 rungs of density. As previously reported, smaller than expected MSP2N2 nanodiscs have been observed if a membrane protein is embedded [31–33]. Yet, the topological flexibility in nanodisc formation has only been observed with SaliPro nanodiscs [44]. Like SaliPro nanodiscs, we saw that the presumed MSP2N2 density is stronger around the membrane protein. This suggests direct interactions between the embedded protein and the scaffolding protein. Again, the observation of others that the nanodisc diameter is altered when a membrane protein is present supports this idea. We suggest that the interaction with the membrane protein enables diverse assemblies and may steer the lipid-to-protein ratios in nanodiscs, to the point that no detergent or lipids are necessary for the nanodisc formation as seen in [45]. These findings are somewhat in conflict with standardized protocols for MSP-based nanodisc formation [42]. Interestingly, a recent study showed that nanodiscs are flexible in solution up to the point that they may disintegrate [46]. Still, the tighter packing of lipids in smaller nanodiscs may be detrimental to the function of the protein, because the lateral pressure in small nanodiscs is higher [37] resulting in conformational bias depending on the nanodisc [36]. Because we observed no density in any binding sites, we presume it is an apo state of ACRB. Despite that, we observe an asymmetric LTO configuration, where at least one protomer (T) should be stabilized by substrate. The asymmetric LTO state has been observed in narrow MSP nanodiscs with or without substrate, and the binding mode of the substrate in nanodiscs is different from the one seen in a crystal [47]. Curiously, we could identify a particle population in a presumed resting state in (all L) [48]. At this point it is unclear if the nanodisc arrests ACRB in a state where states of a transport cycle are observed or if the nanodisc stabilizes the resting state.

Taken together, this study has shown that our approach is feasible and leads to novel insights. Yet, more studies with more complex protein mixtures, and a wider range of cryo-EM map resolutions may be necessary to determine how robust our approach is.

## Materials and methods

### Protein expression

BL21-Tuner cells (Novagen) were transformed with the human Zinc activated ion channel sequence in pET22b (pET22b was a gift from EMBL protein production core). For expression, cells were grown in TB medium at 37°C until $OD_{600}$ of 0.72. Then the culture was pelleted at 6000g, and the medium was exchanged to TB media supplemented with 250 mM sorbitol and equilibrated at 18°C for 45 mins. At an $OD_{600}$ of 0.75 expression was induced with 0.5 mM IPTG at 20° C overnight.

### Solubilisation and enrichment

Cells were resuspended (50 mM Tris, pH 7.2 at RT, 500 mM NaCl, 5% glycerol) and cOmplete protease inhibitor cocktail tablets (Roche) were added. For lysis 1 µL per mL of 100 mg/ml lysozyme (Sigma) was added and incubated at 4°C for 1h. Cells were lysed by sonication (Thermo Fisher Scientific) 50% amplitude, 10 mins, 5 s ON; 15 s OFF. Debris was removed by centrifugation at 10000 g for 30 mins. Membranes were harvested at 108000 g for 1 h, 4C. The membranes were homogenised in 50 mM HEPES, 900 mM KCl, 5% glycerol and one cOmplete protease inhibitor cocktail tablet. Membranes were solubilized with 1% n-Dodecyl β-D-maltoside (DDM, Anatrace)/0.2% Cholesterol Hemi Succinate (CHS, Sigma) 3–4 h at 23°C. Non-solubilized material was removed at 108000g for 30 mins at 4°C. The supernatant was incubated with Ni-NTA resin(G-Bioscience) overnight at 4°C. The resin was washed with homogenisation buffer with w/v 1%DDM/0.2%CHS containing 10 mM imidazole. The protein was eluted in fractions with the Wash buffer and 250–300 mM imidazole.

## MSP encapsulation and size exclusion

The elution fractions were pooled, and the buffer was exchanged to 50 mM HEPES (pH = 7.5), 300 mM KCl w/v 1%DDM/0.2%CHS with PD10 columns (Cytiva) according to manufacturer's recommendations. After protein concentration determination ($OD_{280}$) 1:65 (protein to lipid molar concentration) of Soy Polar lipids (Avanti) were added and incubated for 30 min. Then 1:5 (pentameric protein to MSP molar concentration) of MSP2N2 was added and incubate for 30–40 mins. 50 mg of wet/activated S2 biobeads (BioRad) were added and incubated for 3–4 h. If detergent was not completely removed, another 50 mg of biobeads were added and incubated 3 h or overnight. After detergent removal biobeads were removed and the remaining solution was incubated with NiNTA (G-Biosciences) for 1.5–3 h to remove empty nanodiscs. The beads were washed (50 mM HEPES (pH = 7.5), 300 mM KCl and 10 mM imidazole) and the proteins eluted (50 mM HEPES (pH = 7.5), 300 mM KCl and 250 mM imidazole). The pooled fractions were concentrated and subjected to SEC (ÄktaExplorer, Cytiva) with a Superose 6 Increase 10/300 GL (Cytiva) column, equilibrated with the SEC buffer: 50 mM HEPES (pH = 7.5), 150 mM NaCl and 5 mM EGTA.

## Cryo-EM sample preparation and cryo-EM data collection

SEC Fractions were pooled and concentrated with a Vivaspin® 500 Centrifugal Concentrator MW cut-off 100kD (Sartorius) to 4.6 mg/ml. Quantifoil 2/2 Cu 300 grids were glow discharged for 60 s at 20 mA and 3 µl of the sample was applied to the grids. The grids were blotted for 7 s, at 6°C, 100% humidity with a Vitrobot Mark IV (Thermo Fisher Scientific). The sample was imaged on a Krios (Thermo Fisher Scientific) equipped with a K3 camera in super resolution mode (Gatan). The movies were collected at a nominal magnification of 130,000X (0.66 Å/pixel) and a total dose of 50 e$^-$/Å$^2$ over 40 frames. The defocus was varied between −1 and −2.8 µm. Data were collected using the EPU software (Thermo Fisher Scientific).

## Cryo-EM single particle analysis

All images were processed using CryoSPARC version 4.1–4.5 (Structura Biotechnology Inc., Toronto, ON, Canada) [24]. The raw movies were motion-corrected using patch motion correction. The contrast transfer function (CTF) was estimated using the software's patch CTF estimation. The micrographs were curated based on the CTF fit resolution, motion distance, and ice quality. Particles were manually selected from a small subset of micrographs and classified in 2D. 2D classes with a discernible shape were used as coarse templates for picking. The particles were downsampled and subjected to multiple rounds of 2D classification. 2D classes that shared similarity were selected and further classified. The best 2D classes were used for *Ab initio* map generation. The obtained maps were refined with Non-Uniform refinement. To pick protein-specific particle set we used selected 2D classes from the reconstructed protein as templates. Due to the small number of particles for ARNC, we used the best particles set as a training set for Topaz [49] to pick particles. All newly picked particles were again curated with 2D classification. The remaining particles were further sorted using *Ab Initio* map generation. After refinement, particles were motion corrected before the final Non-Uniform refinement [25]. For ACRB, we used one of the asymmetric volumes from the 3DVA as an input volume for the refinement, due to pseudo-symmetry of the ACRB trimer at low resolution, resulting in poorly resolved protomers, even with C1 symmetry applied. For ARNC, local refinement with a mask including only the ARNC density was performed. To control for template picking bias, circular, featureless templates with 80–180 Å diameter were used for picking and the 2D classes were analysed for ACRB, BO3 and ARNC particles. The reconstruction scheme can be found in S3-S5 Figs.

## 3DVA and refinement

For CryoSPARC 3DVA [10], the refined particles expanded symmetrically (C3) for ACRB and not expanded for BO3. The focusing mask was based on a model we created. For visualisation the volumes we generated in 'cluster mode'.

 

## Unsupervised model building with ModelAngelo

After Non-Uniform Refinement with CryoSPARC the unsharpened half-maps were sharpened in Relion5 [8] with the post-processing tool. To perform the HMMer search [20] we downloaded the *E. coli* proteome (https://www.uniprot.org/proteomes/UP001285702) in fasta format. ModelAngelo was performed in Relion5 with the default parameters. For assessment of the prediction, we inspected the output. The E-value, a probability of false positive residues for the modelled fragment, was chosen as the most important parameter for the quality of the search. All statistics for each map can be found in Tables S3-6 Tables in the supplementary. A closer explanation for the parameter can be found at https://hmmer-web-docs.readthedocs.io/en/latest/searches.html.

## Model building, validation, and structure analysis

For ACRB pdb ID: 2HRT, for BO3 pdb ID: 7N9Z (with lipid densities removed) and for ARNC the AlphaFold prediction of *E. coli* ARNC (P77757) [50] were downloaded and rigidly fitted into the cryo-EM map using ChimeraX version 1.6.1 [51]. Sequences and side chains of the atomic model that were not supported by any density were deleted. The fitted models were refined against the unsharpened cryo-EM map by interactive molecular dynamics flexible fitting (iMDFF) using ISOLDE version 1.6.0 [52] within ChimeraX [51]. A post-processed map generated by deepEMhancer [53] using tightTarget weights was used as a visual aid for interactive model refinement in ISOLDE (but was not used to drive MDFF). We used a strategy outlined in [54]. Each residue was visually inspected at least once. The resulting model for ACRB were submitted for a final real-space refinement using phenix.real_space_refine from the Phenix Suite version 1.20.1–4487 [55] using the settings file generated by ISOLDE. The models for ARNC and BO3 were refined with ServalCat [56]. All model refinement was done against the unsharpened cryo-EM maps. We measured the distance with ChimeraX and by placing markers at the edge of the nanodisc density. All images were generated using ChimeraX.

## In-gel protein digestion and mass spectrometry

Protein bands were excised manually from gels and in-gel digested. Gel pieces were destained following the manufacturer's description. Proteins then were reduced with 0.25 μL of 500 mM dithiothreitol for 45 min at 37°C and alkylated with 0.75 μL of 500 mM iodoacetamide for 30 min at room temperature followed by digestion with 0.5 μg sequencing grade trypsin (Promega) in 50 mM ammonium bicarbonate at 37°C overnight. The tryptic peptides were extracted with 1% formic acid in 2% acetonitrile, followed by 50% acetonitrile twice. The liquid was evaporated to dryness on a vacuum concentrator (Eppendorf).

The reconstituted peptides in solvent A (2% acetonitrile, 0.1% formic acid) were separated on a 50 cm long EASY-spray column (Thermo Fished Scientific) connected to an Ultimate-3000 nano-LC system (Thermo Fisher Scientific) using a 60 min gradient from 4–26% of solvent B (98% acetonitrile, 0.1% formic acid) in 55 min and up to 95% of solvent B in 5 min at a flow rate of 300 nL/min. Mass spectra were acquired on a Q Exactive HF hybrid Orbitrap mass spectrometer (Thermo Fisher Scientific) in m/z 375–1500 at resolution of R = 120000 (at m/z 200) for full mass, followed by data-dependent HCD fragmentations from 17 most intense precursor ions with a charge state 2+ to 7 + . The tandem mass spectra were acquired with a resolution of R = 30000, targeting 2x105 ions, setting isolation width to 1.4 Th and normalized collision energy to 28%.

Acquired raw data files were analysed using the Mascot Server v.2.5.1 (Matrix Science Ltd., UK) and searched against SwissProt protein database with E. coli species selection. Maximum of two missed cleavage sites were allowed for trypsin, while setting the precursor and the fragment ion mass tolerance to 10 ppm and 0.02 Da, respectively. Dynamic modifications of oxidation on methionine, deamidation of asparagine and glutamine and acetylation of N-termini were set. Initial search results were filtered with 5% FDR using Percolator to recalculate Mascot scores. Protein identifications were accepted if they could be established at greater than 95.0% probability and contained at least 2 identified peptides.

## Supporting information

**S1 Fig. SDS Gel in Fig 1.**
(TIF)

**S2 Fig. 'Blob' picking and 2D classification.** A particle set of ~1.5 million was generated with a shapeless circular template (blob). (top) First 2D classification of the generated particle set. Already side views of ACRB (yellow) and BO3 (blue) are recognizable. After a second round of 2D classification with particles without 'junk' (e.g., top, bottom row), classes containing ARNC (purple) become visible.
(TIF)

**S3 Fig. Reconstruction workflow for BO3.** The arrows in the workflow represent an action on the particles. We introduced a second picking step to obtain a more tuned particle set. (bottom right) angular particle distribution and Fourier shell correlation plot for the final particle set in the refinement.
(TIF)

**S4 Fig. Reconstruction workflow for ACRB (top, left) representative micrograph and preliminary templates for picking.** (top right) example for the first 2D classification of the picked particles. The arrows in the workflow represent an action on the particles. (bottom left) angular particle distribution and Fourier shell correlation plot for the final particle set in the refinement.
(TIF)

**S5 Fig. Reconstruction workflow for ARNC.** The arrows in the workflow represent an action on the particles. Due to the small number of particles, we trained a particle picking suite to generate more specific hits. (bottom left) angular particle distribution and Fourier shell correlation plot for the final particle set in the refinement.
(TIF)

**S6 Fig. Structural features of the reconstructed proteins.** (A) sequence comparison of the model generated by ModelAngelo (blue) with the manually modelled/refined AlphaFold model (orange). (B) Helical wheel diagram of the 2 membrane adjacent helices in ARNC. Each represents an amphipathic helix. (C) map density around ubiquinone model taken from the 7N9Z model. (D) map density around cardiolipin model taken from the 7N9Z model (E) additional potential contacts between ACRB and MSP2N2 (W515, and R540) (F) additional contacts between BO3 and MSP2N2 (R455, W545).
(TIF)

**S1 Table. Image processing and model building statistics.**
(XLSX)

**S2 Table. List of identified protein fragments by mass spectrometry.**
(XLSX)

**S3 Table. List of hits for the HHM/ModelAngelo search for the B03 map.** Output from the ModelAngelo run with statistics for each fragment.
(CSV)

**S4 Table. List of hits for the HHM/ModelAngelo search for the ACRB map.** Output from the ModelAngelo run with statistics for each fragment.
(CSV)

**S5 Table. List of hits for the HHM/ModelAngelo search for the ACRB map.** Output from the ModelAngelo run with statistics for each fragment.
(CSV)

**S6 Table. List of the HHM/ModelAngelo statistic for the best hits for each protein.** Compilation of the highest ranked hits for each model that ModelAngelo built and their statistics.
(XLSX)

**S1 Movie. M1 3DVA volume series of ACRB.** Volume series of the 3 LTO clusters found in the 3DVA.
(MP4)

## Acknowledgments

Protein identification was carried out by the Proteomics Biomedicum core facility, Karolinska Institutet (https://ki.se/en/research/proteomics-biomedicum-core-facility). Data were collected at the Cryo-EM Swedish National Facility funded by Knut and Alice Wallenberg, Family Erling Persson, Kempe Foundations, SciLifeLab, Stockholm University, and Umeå University.

## Author contributions

**Conceptualization:** Carsten Mim.

**Data curation:** Qingyang Zhang, Abhinandan Venkatesha Murthy, Carsten Mim.

**Formal analysis:** Qingyang Zhang, Abhinandan Venkatesha Murthy, Carsten Mim.

**Funding acquisition:** Carsten Mim.

**Investigation:** Qingyang Zhang, Abhinandan Venkatesha Murthy.

**Methodology:** Qingyang Zhang, Abhinandan Venkatesha Murthy.

**Visualization:** Carsten Mim.

**Writing – original draft:** Carsten Mim.

**Writing – review & editing:** Qingyang Zhang, Abhinandan Venkatesha Murthy.

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
