## [Decision Letter · Decision Letter 0]

9 Apr 2025

Dear Dr. Mim,

Thank you for submitting your manuscript to PLOS ONE. After careful consideration, we feel that it has merit but does not fully meet PLOS ONE’s publication criteria as it currently stands. Therefore, we invite you to submit a revised version of the manuscript that addresses the points raised during the review process.

We look forward to receiving your revised manuscript.

Kind regards,

Priyanka Sharma

Academic Editor

PLOS ONE

[This study was supported by the Carl Trygger Foundation grant CTS 21:1630 (CM).].

4. Please include a copy of Table 3 which you refer to in your text on page 3.

Additional Editor Comments:

I appreciate the scientific work and the experimental strategies followed to validate your work. I have two following comments:

1. I recommend you to improve the abstract section to convey the nice work you performed in this study.

2. Please use one example and compare your findings with X-ray crystal structure and homology model prediction to enhance the significance of your model.

Thank you!

Reviewers' comments:

Reviewer's Responses to Questions

**Comments to the Author**

1. Is the manuscript technically sound, and do the data support the conclusions?

Reviewer #1: Yes

Reviewer #2: Partly

2. Has the statistical analysis been performed appropriately and rigorously?

Reviewer #1: N/A

Reviewer #2: N/A

3. Have the authors made all data underlying the findings in their manuscript fully available?

Reviewer #1: Yes

Reviewer #2: Yes

4. Is the manuscript presented in an intelligible fashion and written in standard English?

Reviewer #1: Yes

Reviewer #2: Yes

Reviewer #1: In this manuscript, the authors of this work suggested that membrane proteins and membrane scaffolding proteins interact directly. Please give a response to the following:

1) What is the biological relevance of docking scores?

2) The molecular docking results are intriguing, but is there any experimental validation? This necessitates investigations (such as binding assays) to validate any interactions with these proteins.

3) Please provide the gel's figure in the manuscript, but do not provide it as supplemental information.

Reviewer #2: The article “In silico classification and identification co-purified protein complexes yield new structures and multiple MSP assembly states” by Qingyang Zhang, Abhinandan Venkatesha Murthy, and Carsten Mim present a study of the resolution of membrane proteins via cryoEM after the use of nanodiscs and a rather limited purification.

The article basically presents two aspects:

the methodological aspects in getting the models

observations of the interactions of nanodisc scaffolding proteins with the purified proteins.

The first aspect is put forward, including in the title. It is unclear to this reviewer what justifies this choice, the novelty being limited from what is described in the text.

For example, the paper is written in a way that may lead the reader to think that identifying proteins via cryoEM is new, as the authors have a section titled “Unsupervised model building of cryoEM maps can identify proteins”. This is however the results of the paper presenting ModelAngelo. It is unclear if the authors have built further on that because the ways proteins are recognised is not detailed:

lines 120-121 the authors state “Nevertheless, ModelAngelo identified the protein as Undecaprenyl-phosphate 4-deoxy-4- formamido-L-arabinose transferase (ARNC)”

but no details are given and ModelAngelo is not even mentioned in the Methods section

The authors should probably consider changing their wording, including in the title to remove the apparent claims for novelty that are not justified. The observations around the structure of nanodiscs and interaction of scaffold proteins with proteins of interest would probably deserve to be more at the center of the attention of the readers.

**Do you want your identity to be public for this peer review?** For information about this choice, including consent withdrawal, please see our Privacy Policy

Reviewer #1: **Yes:** Mahmoud Balbaa

Reviewer #2: **Yes:** Antoine Taly

---

## [Author Response · Author response to Decision Letter 1]

7 Aug 2025

Review Comments to the Author

We appreciate the time of the reviewers and the critical reading of our manuscript. The comments have strengthened the manuscript, and readability. We highlighted changes to the manuscript in red to simplify the review process.

Reviewer #1: In this manuscript, the authors of this work suggested that membrane proteins and membrane scaffolding proteins interact directly. Please give a response to the following:

1) What is the biological relevance of docking scores?

Thank you for pointing out the weakness in explaining the scoring used. We expanded our explanation for the procedure for the unsupervised model building in the main text. P.4 l.89-91 and p. 6. l.132-134. In addition, we expanded the Methods part to include a description of how we arrived at the ModelAngelo model, p.15 l. 382-390. In short, we are mostly relied on the E-value which is the probability of false positive sequence values for the whole modeled segment/fragment. We included now the output for ModelAngelo and the corresponding values for each fragment in the supporting information (Tables 3-5). As for the finalized model that we generated manually, we first fitted in and then adjusted sidechains etc. The final refinement, meaning the final adjustment of sidechain angles etc. was done with established programs (Phenix, Servalcat), which produced the parameters in Table 1. Therefore, the quality of the initial docking of the model into the cryoEM map is not as important anymore.

2) The molecular docking results are intriguing, but is there any experimental validation? This necessitates investigations (such as binding assays) to validate any interactions with these proteins.

We appreciate the comment and hope to provide some clarification. For ACRB and ARNC only protein models (pdbs) were refined into the cryoEM densities. We have observed low resolution densities representing lipids, but they were far from usable for modelling. For the putative interactions of the membrane protein with the MSP. We tried to reconstruct the cryoEM map for the MSP. However, we never obtained good results that we could use to pinpoint interactions between the membrane protein and the MSP density. Therefore, we refrained from speculating which MSP residues are interacting with the respective membrane protein.

For BO3, we obtained maps that have discernable densities in the position of lipids. Further, we identified densities for the heme groups and well as the ions. The quality of these two groups of non-protein densities was different. For the heme groups the density was well enough defined to correctly assign them. For the Cu and Zn the positioning of these ions were at positions, where we would expect them. Lipid densities are a challenge because we are not sure about their biochemical identity due to the uncertainty of the exact composition of the used soy lipids. So, we refrained from modeling lipids, but referred to findings of others, which seems consistent. However, we modelled and refined the hemes and ions, which is why we used Servalcat. This is the reason why we supplied the fitted lipid densities as supporting information and not deposited these lipids in our model.

3) Please provide the gel's figure in the manuscript, but do not provide it as supplemental information.

The gel image is now new Figure 1, we supplied the gel as a separate image in the supplemental.

Reviewer #2: The article “In silico classification and identification co-purified protein complexes yield new structures and multiple MSP assembly states” by Qingyang Zhang, Abhinandan Venkatesha Murthy, and Carsten Mim present a study of the resolution of membrane proteins via cryoEM after the use of nanodiscs and a rather limited purification.

The article basically presents two aspects:

the methodological aspects in getting the models

observations of the interactions of nanodisc scaffolding proteins with the purified proteins.

The first aspect is put forward, including in the title. It is unclear to this reviewer what justifies this choice, the novelty being limited from what is described in the text.

For example, the paper is written in a way that may lead the reader to think that identifying proteins via cryoEM is new, as the authors have a section titled “Unsupervised model building of cryoEM maps can identify proteins”. This is however the results of the paper presenting ModelAngelo. It is unclear if the authors have built further on that because the ways proteins are recognised is not detailed:

lines 120-121 the authors state “Nevertheless, ModelAngelo identified the protein as Undecaprenyl-phosphate 4-deoxy-4- formamido-L-arabinose transferase (ARNC)” but no details are given and ModelAngelo is not even mentioned in the Methods section The authors should probably consider changing their wording, including in the title to remove the apparent claims for novelty that are not justified. The observations around the structure of nanodiscs and interaction of scaffold proteins with proteins of interest would probably deserve to be more at the center of the attention of the readers.

Thank you for the comment and helping us to avoid any confusion about our contribution.

We modified the title to make it clearer that we are proposing a new workflow. To emphasize this, we are including the information that we are using publicly available software already in the abstract (p.1 l.17-18). We point out in the introduction that other packages exist (p.3 l.58-59) and our reasoning to use ModelAngelo (p.4 l.89-91 and p. 5 l.109). We also expanded on our criteria to assess the hits (p.5 l.126-128) and supplied the output from ModelAngelo in the supporting information (Tables 3-5). We added a separate section in the Methods (p. 15 l.382-390) on the unsupervised model building.

We do believe that our workflow is novel in the sense that we do not require any a priori knowledge about the maps we reconstructed. The ‘build-and-retrieve’ workflow, requires model building and then repicking or mass spectrometry. In fact, our mass spectrometry data provided so many hits, that we still had no idea about the identity of the proteins, which stalled the project. We think that our proof-of-concept will be useful for other scientist that may not have access to a mass spectrometry facility or would like a ‘quick-and-dirty’ overview what may be present in a mixture. The report introducing ModelAngelo demonstrated that it is possible to identify proteins de novo, yet we wanted to test it in a new setting and a specific workflow dedicated to identifying new proteins.

As for the second part regarding the nanodisc assemblies, this was a surprising finding and is not very well documented. It is an emerging realization that membrane scaffold proteins may interact with membrane proteins. Here, we present evidence that this happens in 3 different proteins and that this has an influence on the nanodisc itself. If this is universal needs to be shown.

In conclusion, we believe that both components are a compelling aspect of this study, even if they are somewhat unrelated.

---

## [Decision Letter · Decision Letter 1]

9 Sep 2025

Dear Dr. Mim,

Thank you for submitting your manuscript to PLOS ONE. After careful consideration, we feel that it has merit but does not fully meet PLOS ONE’s publication criteria as it currently stands. Therefore, we invite you to submit a revised version of the manuscript that addresses the points raised during the review process.

We look forward to receiving your revised manuscript.

Kind regards,

Priyanka Sharma

Academic Editor

PLOS ONE

Journal Requirements:

Reviewers' comments:

Reviewer's Responses to Questions

**Comments to the Author**

Reviewer #1: All comments have been addressed

Reviewer #2: All comments have been addressed

Reviewer #3: (No Response)

2. Is the manuscript technically sound, and do the data support the conclusions?

Reviewer #1: Yes

Reviewer #2: Yes

Reviewer #3: Partly

3. Has the statistical analysis been performed appropriately and rigorously?

Reviewer #1: N/A

Reviewer #2: N/A

Reviewer #3: Yes

4. Have the authors made all data underlying the findings in their manuscript fully available?

Reviewer #1: Yes

Reviewer #2: Yes

Reviewer #3: Yes

5. Is the manuscript presented in an intelligible fashion and written in standard English?

Reviewer #1: Yes

Reviewer #2: Yes

Reviewer #3: Yes

Reviewer #1: In response to previous comment 1, the authors primarily rely on the E-value, which is the probability of false positive sequence values across the entire modeled segment/fragment. The supporting information now includes the ModelAngelo output and fragment values (Tables 3-5).

In response to previous ciomment 2, They found low resolution densities representing lipids, but they were far from suitable for modeling. For possible interactions between the membrane protein and the MSP, they attempted to rebuild the cryoEM map for the MSP. They never received satisfactory results that might be used to identify interactions between the membrane protein and the MSP density.

In response to previous comment 3, the authors inserted SDS-PAGE in the manuscript as Figure 1.

No further comments.

Reviewer #2: The authors have properly edited the paper as a response the comment on the initial version of the manuscript.

Reviewer #3: The authors present a workflow to identify proteins from heterogeneous, co-purified membrane-protein preparations. The pipeline leans on using available softwares like ModelAngelo to limit downstream manual refinement. The authors report structures BO3, ACRB, and ARNC which is claimed to be a new structure. They also report unusually small and shape-diverse MSP2N2 nanodiscs and infer direct protein–MSP contacts from density proximity analyses. The study proposed a method to address a topical goal that lowering the barrier to structural proteomics and offers a pragmatic recipe that could be useful. However, several issues should be addressed before the manuscript can be considered for publication.

Major issues

1. The authors mixed up “ubiquitin” with “ubiquinone” throughout the manuscript. This is a major scientific error. Ubiquitin is a protein and ubiquinone, or coenzyme Q is a ligand.

2. In the abstract the author claims that the approach “we only require electron micrographs and a computer for unsupervised model building and protein identification…”. However, identifications are ultimately verified by MS so this claim is a bit too strong in my opinion.

3. The approach seems work well for <4 Å maps and proteomes with good coverage in ModelAngelo/HMMer. Please discuss failure modes (lower resolution, low abundance targets, heavy PTMs/unknown ligands, etc) and when other evidence like MS becomes essential.

4. The authors claim that “one MSP2N2 molecule is enough to form a nanodisc”. This seems to be based only on the diameter estimates and the appearance of “rungs” in the belt density. Please either provide stronger evidence, soften the claim or discuss alternative models at minimum. Simiarly, I think the evidence for “direct interactions between the embedded protein and the scaffolding protein” is not strong enough. Without an atomic MSP model or orthogonal evidence, “direct interaction” should be framed as a hypothesis.

5. For model building and confidence, it would be helpful to add a compact, target-by-target summary of ModelAngelo/HMMER scores (E-value, bit score, bias) and a clear rule set for accepting IDs. Where models are guided by AlphaFold or PDB homologs, explicitly label those regions as guided rather than unsupervised. A residue-wise confidence plot (e.g., per-residue map CC or Q-scores) for the final models would be valuable.

6. The reported diameters vs. a 165 Å “ideal” MSP2N2 disc are central to the conclusions. I think the analysis is not enough here. Please provide a quantitative analysis like distributions from 2D classes and/or map segmentations, measurement methodology, uncertainty estimates, and whether per-view anisotropy or masking could bias diameters measurements.

Minor issues

1. I have noticed several grammar issues and typos. I suggest the authors should carefully proofread the manuscript again. For example,

(1) Line 132. An extra “a” in “We did not supply an individual a sequence, …”.

(2) Line 256. “Solving the structures native proteins…” should be “Solving the structures of native proteins…”

(3) Line 342. A missing “and” in “The pooled fractions were concentrated subjected to SEC…”

(4) Line 374. “…, resulting poorly resolved…” should be “… resulting in poorly resolved…”

(5) Line 397. “BO03” should be “BO3”.

2. Also, some nomenclatures should be consistent throughout. For example, mixed usages of (1) cryoEM/cryo-EM, E. coli (with space)/E.coli (without space), And explain what are hZANC (human Zinc activated ion channel?) and MSP2N2 (membrane-scaffold-protein 2N2?).

**Do you want your identity to be public for this peer review?** For information about this choice, including consent withdrawal, please see our Privacy Policy

Reviewer #1: **Yes:** Mahmoud Esmat Balbaa

Reviewer #2: **Yes:** Antoine Taly

Reviewer #3: No

---

## [Author Response · Author response to Decision Letter 2]

30 Oct 2025

We thank Reviewer #3 for the time to comment on the manuscript. The suggestions made the manuscript stronger. Below we respond the each point.

1. The authors mixed up “ubiquitin” with “ubiquinone” throughout the manuscript. This is a major scientific error. Ubiquitin is a protein and ubiquinone, or coenzyme Q is a ligand.

We thank the author for making us aware of this major mistake. We replaced all instances of "ubiquitin"

2. In the abstract the author claims that the approach “we only require electron micrographs and a computer for unsupervised model building and protein identification…”. However, identifications are ultimately verified by MS so this claim is a bit too strong in my opinion.

We appreciate the suggestion and changed the text accordingly. We emphasized the tentative nature of our approach (l.19-20) and made clear that we used MS to validate our workflow. We also added that we used MS in the introduction and the discussion (l. 66, l. 160).

3. The approach seems work well for <4 Å maps and proteomes with good coverage in ModelAngelo/HMMer. Please discuss failure modes (lower resolution, low abundance targets, heavy PTMs/unknown ligands, etc) and when other evidence like MS becomes essential.

We agree that our approach is promising but has the pitfalls pointed out by the reviewer. We noted that the ARNC was assigned correctly, however due to the low resolution the connectivity and some sidechain assignments are wrong (l. 155-157). ARNC is a relatively small protein, but there were enough fragments to identify the protein. For the first revision, we hinted at these problems (l. 278-280). We followed the reviewer’s suggestion and added more disclaimers (l. 283-288). This includes our case where we saw non-protein densities in the BO3 structure but could not assign them.

4. The authors claim that “one MSP2N2 molecule is enough to form a nanodisc”. This seems to be based only on the diameter estimates and the appearance of “rungs” in the belt density. Please either provide stronger evidence, soften the claim or discuss alternative models at minimum. Similarly, I think the evidence for “direct interactions between the embedded protein and the scaffolding protein” is not strong enough. Without an atomic MSP model or orthogonal evidence, “direct interaction” should be framed as a hypothesis.

We agree with the reviewer that our writing was too enthusiastic. We formulated our findings more neutral. We changed the heading of the paragraph (l. 176-178). We also made it clearer that it is our interpretation that density represents MSP(l. 194-197, legend Figure 4). See also point 6.

5. For model building and confidence, it would be helpful to add a compact, target-by-target summary of ModelAngelo/HMMER scores (E-value, bit score, bias) and a clear rule set for accepting IDs. Where models are guided by AlphaFold or PDB homologs, explicitly label those regions as guided rather than unsupervised. A residue-wise confidence plot (e.g., per-residue map CC or Q-scores) for the final models would be valuable.

We are thankful for the comment and the opportunity to clarify. We added a new table (S6 Table) that lists all targets only. We have no clear cut off for the E-value or any of the other scores. We could imagine that these thresholds are variable, when comparing two isoforms or more broadly two protein family members. Visual inspection of key residue density would be more helpful. The unsupervised model building via ModelAngelo served as an identification tool only. The models were either built based on a pdb template (BO3 and ACRB) or an AlphaFold model. We clarified that in the text (l. 140-141, 147 and 153) and in the figure legend for Figure 3 (l. 172-174) and the Methods part. To address the model quality, we expanded the S1 Table with a section only dealing with the model quality, including the Q score, Atom inclusion and the Map-Model CC. The submitted validation reports contain per sidechain metrics for each of the models. For transparency, we are uploading the original movies to EMPIAR (deposition ID 47487413).

6. The reported diameters vs. a 165 Å “ideal” MSP2N2 disc are central to the conclusions. I think the analysis is not enough here. Please provide a quantitative analysis like distributions from 2D classes and/or map segmentations, measurement methodology, uncertainty estimates, and whether per-view anisotropy or masking could bias diameters measurements.

The reviewer is right that a more precise description would be helpful. We estimated the width of the nanodisc manually. We added this information to the main text (l. 184), figure (l. 211-212) and methods section (l. 434-435). We believe it is more reliable than measuring 2D classes, where tilted projections can distort the distance measurements. The reconstruction mask did encompass the whole nanodisc. We added references describing the size of empty nanodisc. We also searched for cryo-EM maps that encapsulated membrane proteins with MSP2N2. In three cases, the nanodisc was between 10-12nm and therefore smaller than an empty MSP2N2 nanodisc. This finding supports the idea that nanodiscs assembly may be influenced by membrane proteins. A recent paper showed that nanodiscs are dynamic in solution, to the point of near disintegration. We added these references into the main text.

Minor issues

1. I have noticed several grammar issues and typos. I suggest the authors should carefully proofread the manuscript again. For example,

(1) Line 132. An extra “a” in “We did not supply an individual a sequence, …”.

(2) Line 256. “Solving the structures native proteins…” should be “Solving the structures of native proteins…”

(3) Line 342. A missing “and” in “The pooled fractions were concentrated subjected to SEC…”

(4) Line 374. “…, resulting poorly resolved…” should be “… resulting in poorly resolved…”

(5) Line 397. “BO03” should be “BO3”.

We thank the reviewer for noting the errors and understand it impacts the reading. We addressed these and other errors throughout the manuscript.

2. Also, some nomenclatures should be consistent throughout. For example, mixed usages of (1) cryoEM/cryo-EM, E. coli (with space)/E.coli (without space), And explain what are hZANC (human Zinc activated ion channel?) and MSP2N2 (membrane-scaffold-protein 2N2?).

Again, we apologize for the inconsistencies and read the manuscript carefully to correct them.

---

## [Decision Letter · Decision Letter 2]

18 Dec 2025

Exploration of a workflow for the classification and identification of co-purified protein complexes yields new structures and multiple MSP assembly states

PONE-D-25-02134R2

Dear Authors,

We’re pleased to inform you that your manuscript has been judged scientifically suitable for publication and will be formally accepted for publication once it meets all outstanding technical requirements.

Kind regards,

Priyanka Sharma

Academic Editor

PLOS One

Additional Editor Comments (optional):

Reviewers' comments:

Reviewer's Responses to Questions

**Comments to the Author**

Reviewer #3: All comments have been addressed

2. Is the manuscript technically sound, and do the data support the conclusions?

Reviewer #3: Yes

3. Has the statistical analysis been performed appropriately and rigorously?

Reviewer #3: Yes

4. Have the authors made all data underlying the findings in their manuscript fully available?

Reviewer #3: Yes

5. Is the manuscript presented in an intelligible fashion and written in standard English?

Reviewer #3: Yes

Reviewer #3: The authors have addressed all the comments raised in my previous review. I recommend accepting this manuscript.

**Do you want your identity to be public for this peer review?** For information about this choice, including consent withdrawal, please see our Privacy Policy

Reviewer #3: No

---

## [Editor Report · Acceptance letter]

PONE-D-25-02134R2

PLOS One

Dear Dr. Mim,

I'm pleased to inform you that your manuscript has been deemed suitable for publication in PLOS One. Congratulations! Your manuscript is now being handed over to our production team.

Kind regards,

on behalf of

Dr. Priyanka Sharma

Academic Editor

PLOS One